# Selected Methods of Resistance Training for Prevention and Treatment of Sarcopenia

**DOI:** 10.3390/cells11091389

**Published:** 2022-04-20

**Authors:** Tomohiro Yasuda

**Affiliations:** School of Nursing, Seirei Christopher University, Hamamatsu, Shizuoka 433-8558, Japan; tomohiro-y@seirei.ac.jp; Tel.: +81-53-439-1417

**Keywords:** resistance training, sarcopenia, mTOR, high load, low load, blood flow restriction, slow movement and tonic force generation, volitional failure

## Abstract

Resistance training is an extremely beneficial intervention to prevent and treat sarcopenia. In general, traditional high-load resistance training improves skeletal muscle morphology and strength, but this method is impractical and may even reduce arterial compliance by about 20% in aged adults. Thus, the progression of resistance training methods for improving the strength and morphology of muscles without applying a high load is essential. Over the past two decades, various resistance training methods that can improve skeletal muscle mass and muscle function without using high loads have attracted attention, and their training effects, molecular mechanisms, and safety have been reported. The present study focuses on the relationship between exercise load/intensity, training effects, and physiological mechanisms as well as the safety of various types of resistance training that have attracted attention as a measure against sarcopenia. At present, there is much research evidence that blood-flow-restricted low-load resistance training (20–30% of one repetition maximum (1RM)) has been reported as a sarcopenia countermeasure in older adults. Therefore, this training method may be particularly effective in preventing sarcopenia.

## 1. Introduction

With the increasing longevity and decreasing birth rates worldwide, the proportion of elderly individuals in advanced economies has increased dramatically over the last seven decades. In particular, the estimated number of people aged 65 or older in Japan increased to 28.4% in 2019, the highest among 201 countries and regions worldwide [1]. As the national health care costs are high due to the increasing number of older people, the Japanese government has identified solving the health problems associated with the aging of the population as an important issue. In addition, due to changes in the national health care system, more older people are expected to be certified as requiring long-term care, and the number of bedridden aged people is expected to increase. Thus, problems related to the increase in the older population are becoming more serious, and, in the future, the number of physically inactive elderly people who are even more dependent on the medical system is expected to increase.

Sarcopenia, age-related skeletal muscle atrophy, is one of the main causes of physical disability in aged people. Approximately 30% of skeletal muscle mass is lost from 50 to 80 years old [2]. Moreover, denervation of motor units mainly occurs after the age of 70, and other mechanisms, such as hormones, immunity, metabolism, and nutrition, are involved in sarcopenia [3]. Furthermore, previous studies have revealed that sarcopenia is muscle-specific, and elderly people induce significant quadriceps muscle atrophy [4,5]. In other words, sarcopenia is one of the most important factors that reduce the quality of life of the elderly and is associated with morbidity and mortality.

Resistance training improves muscle strength and muscle mass and contributes significantly to the prevention and treatment of sarcopenia, as it is the most essential factor in improving and maintaining sports performance and physical fitness, independent of race, gender, and age [6,7]. Traditionally, high-intensity resistance training using heavy loads increases skeletal muscle size and strength [6]. However, this method has been reported to decrease carotid compliance in healthy people [8], and high mechanical stress must be avoided in patients with bone and joint impairments [9]. Therefore, this training with H-RT is not practical, especially for the elderly and those with low physical fitness, due to the high risk to the cardiovascular and musculoskeletal systems. For this reason, the progression of resistance training methods for increasing the muscle mass and strength without applying a high load is essential. Over the past two decades, various resistance training methods that can induce increases in muscle function/size without the use of high loads have attracted attention, and the studies of their training effects, molecular mechanisms, and safety have been explored.

The present study focuses on the relationship between exercise load/intensity, training effects, and physiological mechanisms as well as the safety of various types of resistance training that have attracted attention as a measure against sarcopenia.

## 2. High-Load Resistance Training (H-RT)

### 2.1. Effects of H-RT

Many studies have shown that resistance training plays an important role in protecting against the effects of sarcopenia [10,11,12]. Several societies, including the National Strength and Conditioning Association (NCSA) and the American College of Sports Medicine (ACSM), have published guidelines to maintain and improve muscle strength, recommending a combination of training frequency, volume, and intensity to achieve optimal strength and muscle mass gains [13,14]. Depending on the individual’s fitness level and training experience, the effectiveness of the required training adaptation is proportional to the resistance training stimulus. In general, the load and intensity of resistance training required to improve muscle strength and achieve skeletal muscle hypertrophy is regarded as 65% or more of the one repetition maximum (1RM) [15,16]. On the other hand, resistance training of less than 65–70% of the 1RM was believed to cause little to no muscle strength gains or skeletal muscle hypertrophy in both young and aged people [17,18] (Figure 1A). Thus, high-load resistance training (H-RT) can be considered an effective countermeasure to sarcopenia (Table 1).

Previous studies have shown that acute/chronic H-RT is a useful preventive intervention against the impact of sarcopenia, and resistance training studies have shown that it increases both muscle protein synthesis (MPS) and skeletal muscle mass in both young and aged adults [31,32,33,34]. Mostly with experiments in young adults and rats, acute H-RT exercise has shown that the mammalian target of the rapamycin (mTOR) signaling pathway [35,36,37,38], including phosphorylation of mTOR and the successive phosphorylation of p70 ribosomal S6 kinase (p70s6K)/rpS6 [36,37,38,39,40] plays an essential role in MPS and muscle hypertrophy. It also appears very likely that phosphorylation of mTOR’s upstream effector, phosphoinositide 3-kinase (PI3K)/Akt [35,36,41], and its downstream effector, 4E-binding protein 1 (4EBP1) [35,36,37,40,41,42], are promoted. Although some studies have reported on the importance of mTOR signaling in the older population, research findings are still very limited [41] (Table 2).

### 2.2. Safety of H-RT 

There is no doubt that H-RT causes muscle hypertrophy in the elderly [51,52,53], but it is very difficult to perform H-RT, especially in the elderly, due to comorbidities such as musculoskeletal disease, coronary artery disease, diabetes [54,55,56], etc. In addition, H-RT is known to cause joint pain due to high compressive loads [57,58], and low-to-moderate-load resistance training is recommended [59,60]. Thus, although resistance training is a useful intervention to protect against the effects of sarcopenia, the number of older adults who can perform H-RT is likely to be quite limited (Table 3).

### 2.3. Perspective of H-RT 

Traditional H-RT for muscle adaptation with resistance exercise is especially difficult for the elderly and patients. Therefore, many studies suggested that the development of resistance training methods that can greatly reduce mechanical stress is critical for safer and more effective muscle hypertrophy and strength promotion for the prevention of sarcopenia. 

## 3. Low-Load Resistance Training with Blood Flow Restriction (L-BFR)

### 3.1. Effects of L-BFR

Low-load resistance training (20–30% 1RM) in combination with blood flow restriction (L-BFR) by an elastic designed cuff belt was originally developed in Japan in the last two decades [68,69], and it is known as KAATSU training [70] (Figure 1B). Most studies investigating L-BFR using weight machines or free weights have established that it leads to increased muscle size and strength [61,69,70,71,72]. L-BFR differs from hemostasis by the use of tourniquets, which completely stop both arteries and veins, by moderately restricting blood flow with specially designed elastic cuff belts while pooling blood in the upper or lower extremities. The effects of L-BFR are thought to be mediated mainly by the following processes: (1) an increase in growth hormone [69], (2) an increase in muscular cell swelling [71,73], and (3) the additional recruitment of fast-twitch muscle fibers [71,72,74,75] due to intramuscular accumulation of the metabolic by-products of L-BFR. These studies implied that the muscle hypertrophic effect of resistance exercise is due to other factors in addition to the large mechanical stress. 

Many training studies on L-BFR have been reported [19,20,21,22,23,24,25], and some studies have compared L-BFR with H-RT in terms of a systematic review and meta-analysis [9,76]. Reviews on adults and older adults have reported that L-BFR can induce similar muscle mass gains when compared to H-RT but has a lower effect on muscle strength. Regarding these effects, Lixandrão et al. [76] showed in adults in the general population that they were independent of blood flow restriction intensity, belt width, and blood flow restriction prescription, while Centner et al. [9] pointed out in older adults that there is still a lack of evidence regarding sex differences, blood flow restriction, and training volume and frequency. Thus, L-BFR induces muscle hypertrophy similar to H-RT, regardless of age [9,77]. 

There are no studies comparing L-BFR and H-RT in the mTOR signaling pathway, but there are acute experiments focusing on L-BFR (one in young adults [46] and one in the elderly [43]). The study in the elderly reported a significant increase in MPS along with a significant increase in Akt phosphorylation, mTOR phosphorylation, and p70S6K phosphorylation after 3 h of L-BFR exercise (20% 1RM) [46]. These results, by the same technique in the same laboratory, are similar to H-RT in young men [37]. Therefore, although the results of L-BFR and H-RT cannot be directly compared, it can be speculated that the activation of the mTOR signaling pathway in L-BFR is comparable to that of H-RT (Table 2).

### 3.2. Safety of L-BFR

Previous cross-sectional studies have found that people who regularly perform H-RT have lower levels of arterial compliance than those who are sedentary [8,78]. On the other hand, our findings indicate that arterial stiffness is maintained after 12 weeks of L-BFR in the elderly. This result is consistent with prior observations that arterial stiffness is not affected after L-BFR in young adults [79]. Although the physiological mechanisms explaining the decrease in arterial compliance after resistance training have not been elucidated, previous L-BFR and H-RT studies have shown that the decrease in arterial compliance due to resistance training is related to an increase in systolic arterial pressure during training sessions [79,80]. Thus, the amount of change in blood pressure response during L-BFR exercise may be a factor affecting arterial compliance with resistance training in young adults as well as in the elderly.

In a previous study from our laboratory, a national survey was conducted in 2016 to determine the use and safety of L-BFR [81]. As in the previous study [82], symptoms including subcutaneous bleeding, numbness, and dizziness were described, but no serious side effects (paralysis due to nerve compression, pulmonary embolism, etc.) were observed. In addition, no serious symptoms (venous thrombosis, rhabdomyolysis, cerebral infarction, cerebral hemorrhage, pulmonary infarction, etc.) were reported in the 2017 study. Thus, L-BFR, when performed under proper guidance, can provide beneficial resistance training benefits with no severe side effects. L-BFR has been reported to improve factors related to falls, such as physical performance and muscle strength [9], although research results on the risk of falls in the elderly are not yet sufficient [83]. In addition, for older adults with knee osteoarthritis, it has been reported to reduce pain to the joints and increase muscle strength, rather than side effects [84,85]. Therefore, L-BFR is expected to have a low level of side effects such as falls and the exacerbation of knee osteoarthritis. Besides, its safety and efficacy have been reported in various rehabilitation studies in middle-aged and older adults (cardiac rehabilitation [86], clinical musculoskeletal rehabilitation [87], medial femoral condylar osteonecrosis cases [88], meniscectomy [89], Churg Strauss syndrome [90], etc.).

### 3.3. Perspective of L-BFR

L-BFR using elastic band exercises is known to produce muscle strength and muscle hypertrophy in older adults [22,23], and various usage methods can be expected [81]. Most studies on L-BFR describe a relatively safe training method without serious side effects in the elderly as well as the young. Therefore, L-BFR is likely to be a very useful intervention to prevent sarcopenia in elderly people and disuse muscle atrophy in elderly people who can only tolerate low-load resistance training (e.g., hip/knee arthritis patients).

## 4. Low-Load Resistance Training with Relatively Slow Movement and Tonic Force Generation (L-ST)

### 4.1. Effects of L-ST

In many cases, L-BFR requires a professional to carefully monitor BFR conditions. Even when the human body is under normal circulation (no blood flow restriction), sustained force generation at 40% maximal voluntary contraction (MVC) has been demonstrated to increase intramuscular pressure in upper [91] and lower [92] extremity muscles, inhibiting both blood inflow and outflow from the muscle. Thus, low-to-moderate intensity resistance exercise (40% MVC) with continuous force generation without the use of an external cuff belt is an effective way to increase muscle strength and size. Subsequent previous studies have shown that low-impact resistance training (L-ST), which generates tonic forces with relatively slow movements, is an available alternative that can be expected to be as effective as L-BFR [93] (Figure 1C). In the L-BFR, muscle blood flow restriction by an external cuff belt causes a decrease in the muscle oxygenation level, an increase in motor unit recruitment, increased blood lactate concentration, and altered endocrine responses/growth factors, suggesting L-BFR-induced skeletal muscle hypertrophy [69,72]. L-ST, on the other hand, is characterized by a relatively slow movement that restricts muscle blood flow and generates a tonic force (3 s in a descending or ascending motion with no relaxation or pause). 

Some studies have examined the relationship between L-ST and muscle size/strength. For example, a previous study reported that the gains in muscle size and strength after L-ST (knee extension, 50% 1RM, three times per week for 12 weeks) were comparable to those after H-RT (80% 1RM) conducted at regular speed (1 s in a descending or ascending motion, 1 s for the relaxation or rest phase) [93]. In addition, L-ST (knee extension, 30% 1RM, twice a week for 12 weeks) resulted in increased strength and muscle hypertrophy in the quadriceps of the elderly [94]. Thus, L-ST increases muscle strength and size not only in the young [93,95] but also in the elderly [26,27,94,96] (Table 1).

Based on findings from studies in young adults, the mechanism of the muscle hypertrophic effect of LST is thought to involve a decrease in intramuscular oxygen concentration due to sustained tension, a corresponding change in the local accumulation of metabolites and the recruitment of muscle fibers [93,95]. Although there is one report of a study in rats that showed that Akt and p70S6K are involved [45], there has not been an investigation of the effect of LST on the mTOR signaling pathway and MPS-induced muscle hypertrophy in humans (Table 2). 

### 4.2. Safety of L-ST

A previous study [95] reported no difference in peak systolic blood pressure during low-intensity (30% 1RM) resistance exercise at L-ST and normal velocity. Therefore, L-ST may be a safe and practical method for older healthy adults [26,27,94,96], older obese adults, and even obese patients suffering from type 2 diabetes [28,29]. Besides, Takenami et al. [29] reported that the arterial stiffness parameters (heart-rate-corrected augmentation index and brachial ankle pulse wave velocity) were unchanged before and after 12 weeks of training. However, there are no reports of studies on the effects of L-ST on fall risk or the exacerbation of knee osteoarthritis in older adults, and the safety and side effects of this training method are not well researched.

### 4.3. Perspective of L-ST

Although L-ST may be a useful method to promote muscle hypertrophy, its safety (e.g., arterial stiffness) and cellular and molecular mechanisms have not yet been fully elucidated by many studies. In addition, Usui et al. [97] reported that L-ST increased task-related muscle strength and skeletal muscle mass but had very little effect on power generation during dynamic explosive exercise. Since L-ST is a relatively slow movement during exercise, it may provide insufficient improvement of the nervous system for fall prevention in aged people. Further studies are needed to understand the effects of L-ST in older adults and evaluate its safety and cellular/molecular mechanisms in inducing MPS and muscle hypertrophy.

## 5. Low-Load Resistance Training until Volitional Failure (L-FAIL)

### 5.1. Effects of L-FAIL

L-BFR studies have revealed that a high load is not a requirement for inducing increased MPS and, ultimately, muscle hypertrophy. Furthermore, similar to the L-BFR observations, it was shown that the same amount of myofiber activation, and a possibly similar stimulation of myofibrillar (MYO) protein synthesis, would occur independent of exercise load if resistance exercise was carried out to volitional fatigue (failure) [35]. Based on these data, it was documented that low-load (30% 1RM) resistance exercise that induces voluntary fatigue (L-FAIL) was found to be more likely to produce stimulation of MPS than H-RT (90% 1RM) or work match resistance exercise in young adults over the past decade [35] (Figure 1C). Additionally, some studies [38,98] reported that L-FAIL induced muscle enlargement similar to that caused by conventional H-RT in healthy young adults. Most studies using L-FAIL have been carried out on young adults, and studies in older adults are very limited, but, in training studies in older adults (three times per week for 12 weeks), L-FAIL (20% 1RM, 80–100 repetitions, one set) was found to induce muscle hypertrophy comparable to H-RT (80% 1RM, 10–15 repetitions, two sets) [30]. 

There are no studies of older adults comparing L-FAIL and H-RT in the mTOR signaling pathway, but there is one study on young adults in acute exercise and one study in chronic exercise. The study on acute exercise [35] reported that elevated PI3K/Akt, elevated mTOR, and elevated 4E-BP1 were found in both conditions, but elevated p70S6K was found only in L-FAIL. In chronic exercise [38], mTOR was found in both conditions, but p70S6K was found only in H-RT. Although studies comparing L-FAIL with H-RT are very limited, in young adults, L-FAIL, like H-RT, activates the muscle protein synthesis pathway via the mTOR signaling pathway. Future research reports on the mTOR signaling pathway in older adults are expected to be published (Table 2). 

### 5.2. Safety of L-FAIL

Studies examining the safety of L-FAIL are limited to studies in young adults. A previous study from our laboratory [64] reported that muscle pain scores (assessed with a visual analogue scale) after L-FAIL were much higher than in a previously reported L-BFR study (20 mm; [62]) (Table 3). In addition, the creatine kinase level (representing muscle damage, *n* = 3) increased gradually, reaching over 11,000 U/L at 96 h post-exercise. These observations indicate that the muscle soreness and muscle damage caused by L-FAIL are similar to those documented in the H-RT study [67] in which participants completed a maximal eccentric exercise (three sets of 10 repetitions) of single elbow flexion. Taken together, the findings suggest that L-FAIL increased muscle swelling at 15 min after exercise, which was primarily due to muscle tissue damage or inflammation. 

In addition, the L-FAIL exercise was only limited to those who were willing to endure the perceptual reaction of the exercise session because the high score on the subjective exertion assessment (15 points or more) was about 1.5 times longer (until spontaneous fatigue) than on the L-BFR.

There are no reports of studies on the effects of L-FAIL on fall risk or the exacerbation of knee osteoarthritis in older adults, and the safety and side effects of this training method are not well researched.

### 5.3. Perspective of L-FAIL

Since L-FAIL-based training methods involve a greater number of repetitions, they require a higher level of motivation over a longer time period than other training exercises. In addition, studies with younger adults have reported high levels of muscle soreness and muscle damage, and these may be induced in older adults as well. Therefore, further research is needed to better understand whether L-FAIL is a training strategy that can be beneficial for older adults.

## 6. Limitations

There are several limitations to this study. First, this study focused on resistance training load intensity and discussed four selected training programs. Therefore, although there have been reports of muscle hypertrophy from training using walking, electrical stimulation, and high-speed power training, this study did not address them. Second, L-ST and L-FAIL were separated. As shown in Figure 1, these two training modalities are visually identical, with the difference being that they are performed with “relatively slow movement and tonic force generation” or “until volitional failure without regard to movement speed”. Since the research backgrounds of each training style are different, this research was designed to examine them separately. 

## 7. Conclusions

Although skeletal muscles in the elderly undoubtedly show sufficient hypertrophy when resistance training is performed under optimal conditions, continuity is very important for resistance training, and training performed as a countermeasure against sarcopenia should ideally include exercises that can be performed throughout an individual’s lifetime. Moreover, it is important to induce sufficient skeletal muscle hypertrophy in the elderly while ensuring safety. From these perspectives, four training programs were discussed in this study. At present, given the existence of many research results, the fact that safety can be ensured, and the fact that it can continue to be performed throughout life, blood-flow-restricted low-load resistance training (20–30% 1RM) is regarded as a particularly effective sarcopenia countermeasure. In the future, it is hoped that more research reports will demonstrate the effectiveness of various resistance training programs as sarcopenia countermeasures.

In this study, resistance training methods with established effects on muscle hypertrophy in older adults were discussed, with a focus on exercise load and exercise intensity. Based on the results of these studies, it is important to determine the resistance training method that is appropriate for each individual and to prevent sarcopenia as early as possible.

## Figures and Tables

**Figure 1 cells-11-01389-f001:**
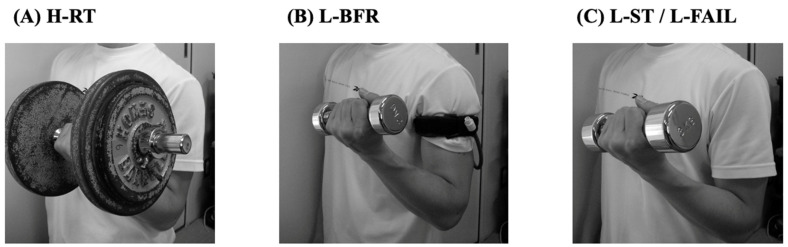
Pictures show a comparison of resistance training of the biceps brachii in each style when using free weights (**A**–**C**). H-RT, high-load (at 70–80% one repetition maximum (1RM)) resistance training. L-BFR, low-load resistance training (at 20–30% 1RM) with blood flow restriction by an elastic designed cuff belt. L-ST, low-load (at 20–30% 1RM) resistance training with relatively slow movement and tonic force generation. L-FAIL, low-load (at 20–30% 1RM) resistance exercise to volitional fatigue.

**Table 1 cells-11-01389-t001:** Characteristics of each type of resistance training for improving skeletal muscle mass in older adults based on previous studies.

Characteristics	Resistance Training			
	H-RT	L-BFR	L-ST	L-FAIL
Load	70–85% 1RM	10–50% 1RM	BW, 30–50% 1RM	20% 1RM
Frequency (day/week)	2–3	2–3	2–7	3
Sets x Repetitions	1–3 x 8–15	1–4 x 15–30	1–3 x 5–15	1 x 80–100
References	[6,14,17,18]	[9,19,20,21,22,23,24,25]	[26,27,28,29]	[30]

The numbers in parentheses are references. 1RM, one repetition maximum. BW, body weight. H-RT, high-load resistance training. L-BFR, low-load resistance training with blood flow restriction by an elastic designed cuff belt. L-FAIL, low-load resistance exercise to volitional fatigue. L-ST, low-load resistance training with relatively slow movement and tonic force generation.

**Table 2 cells-11-01389-t002:** Effects of resistance exercise/training on mTOR signaling pathway in young/older adults or rat models based on previous studies.

Resistance Training												
	H-RT			L-BFR			L-ST			L-FAIL		
	YH	OH	Rat	YH	OH	Rat	YH	OH	Rat	YH	OH	Rat
mTOR signaling pathway												
PI3K/Akt	↑ [35,41]		↑ [36]		↑ [43]	↑ [44]			↑ [45]	↑ [35]		
mTOR	↑ [35,37,38]	↑ [41]	↑ [36]		↑ [43]					↑ [35,38]		
S6K1/p70S6K	↑ [37,38,39,40]		↑ [36]	↑ [46]	↑ [43]	↑ [44,47,48]			↑ [45]	↑ [35]		
rpS6			↑ [36,42]			↑ [49,50]						
4E-BP1	↑ [35,37,40,41]		↑ [36,42]							↑ [35]		

The numbers in parentheses are references. 4E-BP1, 4E-binding protein 1. H-RT, high-load (at 70–80% one repetition maximum (1RM)) resistance training. L-BFR, low-load resistance training with blood flow restriction by an elastic designed cuff belt. L-FAIL, low-load resistance exercise to volitional fatigue. L-ST, low-load resistance training with relatively slow movement and tonic force generation. mTOR, mammalian target of rapamycin. OH, older adult humans. p70, ribosomal protein kinase 1. p70kDa, ribosomal protein S6 kinase. PI3K, phosphatidylinositol-3 kinase. YH, young adult humans. S6K1, p70S6K. rpS6, ribosomal protein S6. ↑: Significant increase.

**Table 3 cells-11-01389-t003:** Benefits and potential complications of different resistance training modes in older adults.

Resistance Training				
	H-RT	L-BFR	L-ST	L-FAIL
Benefits 1	Exercise repetition: Few [6,18,19]	Exercise load: Low [9,19,20,21,22,23,24,25]	Exercise load: Low [26,27]	Exercise load: Low [30]
Benefits 2	Strength gain: Large [9,38]	Arterial stiffness: No change [21,22,23]		
Benefits 3		Versatility: High [9,22,25,61]		
Potential complications 1	Pain in bones and joints: Occurrence [9]	Muscle soreness and damage: Occurrence * [62,63]	Not applicable (Few reports)	Muscle soreness and damage: Occurrence * [64]
Potential complications 2	Arterial stiffness: Increase * [8]	Rhabdomyolysis: Occurrence * [65,66]		Discomfort: Occurrence * [64]
Potential complications 3	Muscle soreness and damage: Occurrence [67]			

The numbers in parentheses are references. * = Reference of young adults. H-RT, high-load (at 70–80% one repetition maximum (1RM)) resistance training. L-BFR, low-load resistance training with blood flow restriction by an elastic designed cuff belt. L-FAIL, low-load resistance exercise to volitional fatigue. L-ST, low-load resistance training with relatively slow movement and tonic force generation.

## Data Availability

Not applicable.

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
