# Peer review of "Selected Methods of Resistance Training for Prevention and Treatment of Sarcopenia"

_cells, 2022, doi:10.3390/cells11091389_

Round 1
Reviewer 1 Report
This is a review of resistance training methods to prevent and treat sarcopenia. The review is well done and written, so I congratulate the author for his work. However, I believe that throughout the review, comparisons should be carried out as directly as possible between traditional resistance training (i.e., H-RT) and all alternative methods suggested in the context of sarcopenia and aging (or populations affected by sarcopenia). In the absence of a direct comparison, the author should indicate this in the topics about perspectives. Finally, gaps in the literature should be better indicated, and possible biases in some statements should be minimized. Below, I present some aspects that I believe need to be reviewed in detail.
Introduction
Line 51-52: Other aspects need to be mentioned.
Line 58-60: The physiological mechanisms related to adaptations were not mentioned in the review's objectives.
2. High-load resistance training (H-RT)
2.1. Effects of H-RT
Line 72-75: Sentence accuracy needs to be improved. What does the author mean by “no substantial”? Furthermore, the statement is partially true, as intensities lower than 65-70% 1RM can maximize adaptations if exercise is performed to or near concentric muscle failure. Finally, the sentence accompanying the indication in figure 1 does not seem to be the most adequate.
Figure 1 and Table 1 should show 20-30%, as most studies with low load without BFR were not performed with 20% of 1RM. Additionally, in Table 1, it is necessary to inform that the % refers to the 1-RM test.
2.2. Cellular and molecular mechanisms…
I understand that this journal has a more mechanistic focus, but the way these topics were developed throughout the review, I think they should not be part of this article. However, if the author understands that they should be kept for a reason mentioned initially, he would like to see a greater depth in each of the mechanistic topics, with more precision of information, clear indications of gaps in the literature, etc.
2.3. Safety of H-RT and 2.4. Perspective of H-RT
This topic should be revised entirely. At various times the author uses acute studies involving the young population and studies with loads above what the author characterizes as training with high loads. In the absence of references, the author should be careful when extrapolating studies that do not directly apply to the context of the review. Finally, safety issues should be expanded to aspects beyond cardiovascular and hemodynamics.
3.1. Effects of L-BFR
Lines 174-182: Review the sentence, as gains are not the same throughout a training program. Regarding the comparison between L-BFR and H-RT, use studies that made this comparison directly. There are excellent meta-analyses that compared these two protocols (PMID: 29043659; PMID: 30306467).
Lines 184-185: The studies cited do not appear in table 1 as indicated in these sentences. In addition, I suggest citing the meta-analyses presented above here.
3.2. Cellular and molecular mechanisms…
Lines 193-197: The sentence should be revised as the studies cited do not make a direct comparison between H-RT and L-BFR. Please indicate this gap in the literature in the perspective section.
3.3. Safety of L-BFR
This topic should be expanded. Many other security-related aspects have already been investigated and should be included in this topic.
3.4. Perspective of L-BFR
Why is walking associated with BFR included in this topic but not others?
4. Low-load resistance training with relatively slow movement…
I think this topic is very close to L-Fail. I believe these two topics can be put together and placed in the context of low load training until (or close to) muscle failure. Increasing the time under tension can be a strategy to anticipate the proximity of muscle failure.
5. Low-load resistance training until volitional failure (L-FAIL)
5.1. Effects of L-FAIL
5.2. Cellular and molecular mechanisms
Lines 303-306. This is a mechanism that still needs to be investigated further. In my view, there is a big leap in logic in relation to what references 88 and 89 show and what is speculated from existing studies with exercises, which use only acute measurements of muscle thickness or echo intensity to support an extremely complex phenomenon.
I recommend that these topics be reviewed and that solid mechanistic evidence be used, particularly to compare L-FAIL with H-RT.
Conclusions and Abstracts should be modified based on changes made to the entire manuscript.
Author Response
Dear Reviewer 1
Thank you very much for reviewing this manuscript and for your constructive and helpful comments. I have responded to the point raised by you, and revised the manuscript where necessary and explained my thoughts in this response letter (Red Letters). I hope that the revisions will satisfy your standard.
As much as possible, revisions have been made to describe studies that directly compare each training with H-RT. Gaps in the references were also revised to minimize bias.
Tomohiro Yasuda
Reviewer: 1
Comments to the Author
This is a review of resistance training methods to prevent and treat sarcopenia. The review is well done and written, so I congratulate the author for his work. However, I believe that throughout the review, comparisons should be carried out as directly as possible between traditional resistance training (i.e., H-RT) and all alternative methods suggested in the context of sarcopenia and aging (or populations affected by sarcopenia). In the absence of a direct comparison, the author should indicate this in the topics about perspectives. Finally, gaps in the literature should be better indicated, and possible biases in some statements should be minimized. Below, I present some aspects that I believe need to be reviewed in detail.
[ANSWER]
Dear Reviewer
Thank you very much for reviewing this manuscript and for your constructive and helpful comments. I
have responded to the point raised by you, and revised the manuscript where necessary and explained my
thoughts in this response letter (Red Letters). I hope that the revisions will satisfy your standard.
As much as possible, revisions have been made to describe studies that directly compare each
training with H-RT. Gaps in the references were also revised to minimize bias.
< 1 >
Introduction
Line 51-52: Other aspects need to be mentioned.
[ANSWER] At the reviewer's suggestion, the text was revised as follows. Reference #62 was changed as #9.
Line 49-53
...However, this method has been reported to decrease carotid compliance in healthy people [8], and high
mechanical stress must be avoided in patients with bone and joint impairments [9]. Therefore, this training with H-RT is not
practical, especially for the elderly and those with low physical fitness, due to the high risk to the cardiovascular and
musculoskeletal systems....
9. Centner, C.; Wiegel, P.; Gollhofer, A.; König, D. Effects of blood flow restriction training on muscular strength and
hypertrophy in older individuals: A systematic review and meta-analysis. Sports Med. 2019, 49, 95-108.
< 2 >
Line 58-60: The physiological mechanisms related to adaptations were not mentioned in the review's objectives.
[ANSWER]
At the suggestion of reviewer, the word "physiological mechanisms" has been added to the purpose as
follows.
Line 15-17(Abstract), Line 59-61 (Introduction)
... The present study focuses on the relationship between exercise load/intensity, training effects, and physiological
mechanisms, as well as the safety of various types of resistance training, which have attracted attention as a measure against
sarcopenia.
< 3 >
2. High-load resistance training (H-RT)
2.1. Effects of H-RT
Line 72-75: Sentence accuracy needs to be improved. What does the author mean by “no substantial”? Furthermore,
the statement is partially true, as intensities lower than 65-70% 1RM can maximize adaptations if exercise is
performed to or near concentric muscle failure. Finally, the sentence accompanying the indication in figure 1 does
not seem to be the most adequate.
[ANSWER] As the reviewer said, I felt that the description was misleading." The word "substantial" has
been removed. And description in figure 1 was revised as below.
Line 75-78
...On the other hand, resistance training of less than 65%-70% 1RM was believed to cause little to no substantial
muscle strength gains or skeletal muscle hypertrophy in both young and aged people [17,18] ...
Figure 1
Pictures show a comparison of resistance training of the biceps brachii in each style when using free weights. ...
< 4 >
Figure 1 and Table 1 should show 20-30%, as most studies with low load without BFR were not performed with
20% of 1RM. Additionally, in Table 1, it is necessary to inform that the % refers to the 1-RM test.
[ANSWER] According to the suggestions of reviewer 1 and reviewer 2, Figure 1 (description) and Table 1
were revised as manuscript.
< 5 >
2.2. Cellular and molecular mechanisms...
I understand that this journal has a more mechanistic focus, but the way these topics were developed throughout
the review, I think they should not be part of this article. However, if the author understands that they should be
kept for a reason mentioned initially, he would like to see a greater depth in each of the mechanistic topics, with
more precision of information, clear indications of gaps in the literature, etc.
[ANSWER] According to the suggestions of reviewer, the mechanism is included in the "Effects of ..."
section. Moreover, the content was compared to that of H-RT as much as possible, including citation of
references and whether the study was conducted on young or older adults. In order to avoid complicating
the description of mechanisms, Table 2 is set to only include mTOR signaling pathways (satellite cell
regulators were deleted).
Line 109-117
Previous studies have shown that acute/chronic H-RT is a useful preventive intervention against the impact of
sarcopenia, and resistance training researches have shown that it increases both muscle protein synthesis (MPS) and skeletal
muscle mass in both young and aged adults [31-34]. Mostly with experiments in young adults and rats, acute H-RT exercise
has shown that the mammalian target of rapamycin (mTOR) signaling pathway [35-38], including phosphorylation of mTOR
and successive phosphorylation of p70 ribosomal S6 kinase (p70s6K)/rpS6 [36-40], plays an essential role in MPS and muscle
hypertrophy. It also appears very likely that phosphorylation of mTOR's upstream effector, phosphoinositide 3-kinase
(PI3K)/Akt [35,38,41], and its downstream effectors, 4E-binding protein 1 (4EBP1) [35,36,38,40-42] are promoted. Although
some studies have reported on the importance of mTOR signaling in the older population, research findings are still very limited
[41] (Table 2).
4
Line 201-209
There are no studies comparing L-BFR and H-RT in the mTOR signaling pathway, but there are acute experiments
focusing on L-BFR (one in young adults [43] and one in the elderly [44]). The study in the elderly reported a significant
increase in MPS along with a significant increase in Akt phosphorylation, mTOR phosphorylation, and p70S6K
phosphorylation after 3 hours of L-BFR exercise (20% 1RM) [43]. These results by the same technique in the same laboratory
are similar to H-RT in young men [28]. Therefore, although the results of L-BFR and H-RT cannot be directly compared, it can
be speculated that the activation of the mTOR signaling pathway in L-BFR is comparable to that of H-RT (Table 2).
Line 274-277
Based on findings from studies in young adults, the mechanism of the muscle hypertrophic effect of LST is thought
to involve a decrease in intramuscular oxygen concentration due to sustained tension, a corresponding change in local
accumulation of metabolites and the recruitment of muscle fibers. [87,89]. Although there is one report of a study in rats that
Akt and p70S6K are involved [50], there has not been investigation of the effect of LST on mTOR signaling pathway and MPS-
induced muscle hypertrophy in humans (Table 2).
Line 318-326
There are no studies of older adults comparing L-FAIL and H-RT in the mTOR signaling pathway, but there is one
study on young adults in acute exercise and one study in chronic exercise. Acute exercise [35] reported that elevated PI3K/Akt,
elevated mTOR, and elevated 4E-BP1 were found in both conditions, but elevated p70S6K was found only in L-FAIL. In chronic
exercise [37], mTOR was found in both conditions, but p70S6K was found only in H-RT. Although studies comparing L-FAIL
with H-RT are very limited, in young adults, L-FAIL, like H-RT, activates the muscle protein synthesis pathway via the mTOR
signaling pathway. Future research reports on the mTOR signaling pathway in the older adults are expected to be published
(Table 2)
2.3. Safety of H-RT and 2.4. Perspective of H-RT
< 6 >
This topic should be revised entirely. At various times the author uses acute studies involving the young population
and studies with loads above what the author characterizes as training with high loads. In the absence of references,
the author should be careful when extrapolating studies that do not directly apply to the context of the review.
Finally, safety issues should be expanded to aspects beyond cardiovascular and hemodynamics.
[ANSWER] According to the suggestions of reviewer, this topic was revised entirely. Safety issues were
expanded to include aspects other than cardiovascular and hemodynamic issues, and content related to the
elderly.
Line 141-147
2.2 Safety of H-RT
There is no doubt that H-RT causes muscle hypertrophy in the elderly [51-53], but it is very difficult to perform H-
RT, especially in the elderly, due to comorbidities such as musculoskeletal disease, coronary artery disease, and diabetes [54-
56], etc.. Thus, although resistance training is a useful intervention to protect against the effects of sarcopenia, the number of
older adults who can perform H-RT is likely to be quite limited (Table 3).
Line 149-154
2.3 Perspective of H-RT
Traditional H-RT for muscle adaptation with resistance exercise is especially difficult for the elderly and patients.
Therefore, many studies suggested that the development of resistance training methods that can greatly reduce mechanical
stress is critical for safer and more effective muscle hypertrophy and strength promotion for the prevention of sarcopenia.
Added references
54. Hoffman, C.; Rice, D.; Sung, H. Persons with chronic conditions: their prevalence and costs. JAMA. 1996, 276,1473–1479.
55. Gheno, R.; Cepparo, J.M.; Rosca, C.E.; Cotton, A. Musculoskeletal disorders in the elderly. J. Clin. Imaging Sci. 2012,
2, 39.
56. Papa, E.V.; Dong, X.; Hassan, M. Skeletal muscle function deficits in the elderly: current perspectives on resistance training.
J. Nat. Sci. 2017, 3, e272.
3.1. Effects of L-BFR
< 7 >
Lines 174-182: Review the sentence, as gains are not the same throughout a training program. Regarding the
comparison between L-BFR and H-RT, use studies that made this comparison directly. There are excellent meta-
analyses that compared these two protocols (PMID: 29043659; PMID: 30306467).
[ANSWER] At the reviewer's suggestion, the text was revised as follows. PMID: 30306467 was already listed
in this study, and PMID: 29043659 was newly added as reference #73.
Lines 191-199
Many training studies on L-BFR have been reported [19-25], and some studies have compared L-BFR with H-RT in terms
of systematic review and meta-analysis [9,72]. Not only reviews on adults, but also reviews on older adults have reported that
L-BFR can induce similar muscle mass gains when compared to H-RT, but has a lower effect on muscle strength. Regarding
these effects, Lixandrão et al. [72] in the adults in general population showed that they were independent of blood flow
restriction intensity, belt width, and blood flow restriction prescription, while Centner et al. [9] in the older adults pointed out
that there is still a lack of evidence regarding sex differences, blood flow restriction, and training volume and frequency.
72. Lixandrão, M.E.; Ugrinowitsch, C.; Berton, R.; Vechin, F.C.; Conceição, M.S.; Damas, F.; Libardi, C.A.; Roschel,
H. Magnitude of muscle strength and mass adaptations between high-load resistance training versus low-load
resistance training associated with blood-flow restriction: A systematic review and meta-analysis. Sports Med.
2018, 48, 361-378.
< 8 >
Lines 184-185: The studies cited do not appear in table 1 as indicated in these sentences. In addition, I suggest
citing the meta-analyses presented above here.
[ANSWER] According to the reviewer's suggestion, the text was revised (deleted the description of “Table
1”) as follows.
Line 199-200
... Thus, L-BFR induces muscle hypertrophy similar to H-RT, regardless of age [9,73].
3.2. Cellular and molecular mechanisms...
< 9 >
Lines 193-197: The sentence should be revised as the studies cited do not make a direct comparison between H-
RT and L-BFR. Please indicate this gap in the literature in the perspective section.
[ANSWER] According to the suggestions of reviewer, the mechanism is included in the "Effects of ..."
section. Moreover, the content was compared to that of H-RT as much as possible, including citation of
references and whether the study was conducted on younger or older adults.
Line 201-209
There are no studies comparing L-BFR and H-RT in the mTOR signaling pathway, but there are acute experiments
focusing on L-BFR (one in young adults [43] and one in the elderly [44]). The study in the elderly reported a significant
increase in MPS along with a significant increase in Akt phosphorylation, mTOR phosphorylation, and p70S6K
phosphorylation after 3 hours of L-BFR exercise (20% 1RM) [43]. These results by the same technique in the same laboratory
are similar to H-RT in young men [36]. Therefore, although the results of L-BFR and H-RT cannot be directly compared, it can
be speculated that the activation of the mTOR signaling pathway in L-BFR is comparable to that of H-RT (Table 2).
3.3. Safety of L-BFR
< 10 >
This topic should be expanded. Many other security-related aspects have already been investigated and should be
included in this topic.
[ANSWER] According to the reviewer's suggestion, it was noted that various cases have also shown
effectiveness and safety as follows.
Line 228-233
... Thus, L-BFR, when performed under proper guidance, can provide beneficial resistance training benefits with
no severe side effects. In addition, its safety and efficacy have been reported in various rehabilitation studies in middle-aged
and older adults (cardiac rehabilitation [79], clinical musculoskeletal rehabilitation [80], medial femoral condylar
osteonecrosis cases [81], knee arthritis [82], meniscectomy [83], and Churg Strauss syndrome [84], etc.).
< 11 >
3.4. Perspective of L-BFR
Why is walking associated with BFR included in this topic but not others?
[ANSWER] It was believed that walking training had nothing to do with muscle hypertrophy. However,
since 2006, it has been reported that "BFR + walking" induces muscle hypertrophy (Abe et al. 2006 J Appl
Physiol, Ozaki et al. 2011 J Gerontology). However, other training styles, such as H-RT and L-ST/L-FAIL,
have not been reported to have a muscle hypertrophy effect using walking. As the reviewer stated, I felt that
mentioning walking in this review would blur the focus of this review, as it would require mentioning the
situation in other topics as well. Therefore, it was decided to remove the mention of walking. However, the
points you raised are important, so we have decided to list them as "limitations."
Line 236-237
L-BFR using walking or elastic band exercises is known to produce muscle strength and muscle hypertrophy
increase in elderly people [21,25],...
Line 352-361
Limitations
8
There are several limitations to this study. First, this study focused on resistance training load intensity and
discussed four training programs. Therefore, although there have been reports of muscle hypertrophy from training using
walking and electrical stimulation, this study did not address them. Second, L-ST and L-FAIL were separated. As shown in
Figure 1, these two training modalities are visually identical, with the difference being that they are performed "relatively slow
movement and tonic force generation " or "until volitional failure without regard to movement speed”. Since the research
backgrounds of each training style are different, this research was designed to examine them separately.
4. Low-load resistance training with relatively slow movement...
< 12 >
I think this topic is very close to L-Fail. I believe these two topics can be put together and placed in the context of
low load training until (or close to) muscle failure. Increasing the time under tension can be a strategy to anticipate
the proximity of muscle failure.
[ANSWER] L-FAIL and LST are similar in terms of "exhaustion”. However, it is also noted that LST is a
training that got its idea from L-BFR. Therefore, the three types of low-intensity training discussed in this
review all have "low-intensity + exhaustion" as their key point.
Since each training has its own research background and results, etc., I would like to categorize and
introduce those three points here as I initially did.
I also show that L-FAIL and LST are the same simple condition (Figure 1), so the reader will understand
that these two types of training are quite close. As a limitation, I also describe the difficulty regarding
whether or not to classify L-BFR and LST as below.
Line 356-361
... Second, L-ST and L-FAIL were separated. As shown in Figure 1, these two training modalities are visually
identical, with the difference being that they are performed "relatively slow movement and tonic force generation " or "until
voli-tional failure without regard to movement speed”. Since the research back-grounds of each training style are different,
this research was designed to ex-amine them separately.
5. Low-load resistance training until volitional failure (L-FAIL)
5.1. Effects of L-FAIL
5.2. Cellular and molecular mechanisms
< 13 >
Lines 303-306. This is a mechanism that still needs to be investigated further. In my view, there is a big leap in
logic in relation to what references 88 and 89 show and what is speculated from existing studies with exercises,
which use only acute measurements of muscle thickness or echo intensity to support an extremely complex
phenomenon. I recommend that these topics be reviewed and that solid mechanistic evidence be used, particularly
to compare L-FAIL with H-RT.
[ANSWER] According to the suggestions of reviewer, the content was compared to that of H-RT as much
as possible, including citation of references and whether the study was conducted on young or older adults.
In order to avoid complicating the description of mechanisms, the evidence is used to only include mTOR
signaling pathways (satellite cell regulators were deleted). In addition, revisions have been done for 5.1, 5.2
and 5.3.
Line 312-326
5.1 Effects of L-FAIL
... Therefore, L-FAIL is likely to be an effective way to promote MPS and muscle enlargement with or without BFR.
Thus, there is a high probability that L-FAIL is an effective way to promote MPS and muscle hypertrophy with or without BFR.
Most studies using L-FAIL have been carried out on young adults, and very limited studies have been reported on older adults.
However, in a training study with older adults (3 times per week for 12 weeks), L-FAIL (20% 1RM, 80-100 repetitions, 1 set)
was found to induce muscle hypertrophy comparable to H-RT (80% 1RM, 10-15 repetitions, 2 sets) [30].
There are no studies of older adults comparing L-FAIL and H-RT in the mTOR signaling pathway, but there is one
study on young adults in acute exercise and one study in chronic exercise. Acute exercise [35] reported that elevated PI3K/Akt,
elevated mTOR, and elevated 4E-BP1 were found in both conditions, but elevated p70S6K was found only in L-FAIL. In chronic
exercise [37], mTOR was found in both conditions, but p70S6K was found only in H-RT. Although studies comparing L-FAIL
with H-RT are very limited, in young adults, L-FAIL, like H-RT, activates the muscle protein synthesis pathway via the mTOR
signaling pathway. Future research reports on the mTOR signaling pathway in the older adults are expected to be published
(Table 2).
Line 328-329
5.2 Safety of L-FAIL
Studies examining the safety of L-FAIL are limited to studies in young adults. A previous study from our laboratory
[63] reported that muscle pain scores ...
Line 344-350
5.3 Perspective of L-FAIL
Since L-FAIL based training methods involve a greater number of repetitions, they require a higher level of
motivation over a longer time period than other training exercises. In addition, studies with younger adults have reported high
levels of muscle soreness and muscle damage, and these may be induced in older adults as well. Therefore, further research is
needed to better understand whether L-FAIL is a training strategy that can be beneficial for older adults.
< 15 >
Conclusions and Abstracts should be modified based on changes made to the entire manuscript.
[ANSWER] According to the reviewer's suggestion, the text was revised as follows.
Line 8-20
Abstract: Resistance training is an extremely beneficial intervention to prevent and treat of sarcopenia. In general, traditional
high-intensity resistance training improves skeletal muscle morphology and strength, but this method is impractical and may
even reduce arterial compliance by about 20% in aged adults. Thus, the progression of resistance training methods for
improving the strength and morphology of muscles without applying a high load is essential. Over the past two decades, various
resistance training methods that can improve skeletal muscle mass and muscle function without using high loads have attracted
attention, and their training effects, molecular mechanisms, and safety have been reported. The present study focuses on the
relationship between exercise load/intensity, training effects, and physiological mechanisms, as well as the safety of various
types of resistance training, which have attracted attention as a measure against sarcopenia. At present, there is much research
evidence that blood flow-restricted low-load resistance training (20%–30% of one repetition maximum [1RM]) has been
reported as a sarcopenia countermeasure in older adults. Therefore, this training method may be particularly effective in
preventing sarcopenia.
Line 369-380
Conclusions
Although skeletal muscles in the elderly undoubtedly show sufficient hypertrophy when resistance training is
performed under optimal conditions, continuity is very important for resistance training, and training performed as a
countermeasure against sarcopenia should ideally include exercises that can be performed throughout an individual’s lifetime.
Moreover, it is important to induce sufficient skeletal muscle hypertrophy in the elderly while ensuring safety. From these
perspectives, four training programs were discussed in this study. At present, given the existence of many research results, the
fact that safety can be ensured, and the fact that it can continue to be performed lifelong, blood flow-restricted low-load
resistance training (20-30% 1RM) is regarded as particularly effective as a sarcopenia countermeasure. In the future, it is
hoped that more research reports will demonstrate the effectiveness of various resistance training programs as a sarcopenia
countermeasure.
In this study, resistance training methods with established effects on muscle hypertrophy in the older adults were
discussed with a focus on exercise load and exercise intensity. Based on the results of these studies, it is important to determine
the resistance training method that is appropriate for each individual and to prevent sarcopenia as early as possible.
Reviewer 2 Report
Review
Various methods of resistance training for prevention and treatment of sarcopenia
In this review paper the Author describes some methods of resistance exercises in the context of prevention and treatment of sarcopenia.
Though potentially interesting, the manuscript seems bit selective and somewhat chaotic. Several potential training modes have not been described (power training, electrostimulation).
Described side-effects are mainly limited to arterial compliance/stiffness. Other potential complications (cardiovascular, injuries) should be systematically presented.
I would suggest presenting in one table benefits and potential complications of different resistance training modes in older adults.
Specific comments
Abstract
“…reduce the function of the cardiovascular system…” – not clear.
Last sentence is too long.
Figure 1. Explain the abbreviations.
Table 1. Provide references.
Table 2. Explain the abbreviations.
Line 137 and elsewhere: multiple sclerosis is not an epidemiologic problem of older adults
Lines 139/140 and 143/144: redundant repetition
Lines 184/185, 265, 314: Table 1 shows something different
Line 304: Citations do not correspond to the text
Author Response
Reviewer: 2
Comments to the Author
Various methods of resistance training for prevention and treatment of sarcopenia
In this review paper the Author describes some methods of resistance exercises in the context of prevention and
treatment of sarcopenia.
Though potentially interesting, the manuscript seems bit selective and somewhat chaotic. Several potential training
modes have not been described (power training, electrostimulation).
Described side-effects are mainly limited to arterial compliance/stiffness. Other potential complications
(cardiovascular, injuries) should be systematically presented.
I would suggest presenting in one table benefits and potential complications of different resistance training modes
in older adults.
[ANSWER]
Dear Reviewer
Thank you very much for reviewing this manuscript and for your constructive and helpful comments. I
have responded to the point raised by you, and revised the manuscript where necessary and explained my
thoughts in this response letter (Red Letters). I hope that the revisions will satisfy your standard.
Because the review focused on external loading, it did not report on training such as electrical stimulation.
This was noted in the "Limitations" section. In addition, the benefits and potential complications of various
types of training are also summarized in Table 3 as manuscript.
< 1 >
Specific comments
Abstract
2
“...reduce the function of the cardiovascular system...” – not clear.
[ANSWER] At the reviewer's suggestion, the text was revised as follows for greater clarity.
Line 10
... but this method is impractical and may even reduce arterial compliance by about 20% in aged adults.
< 2 >
Last sentence is too long.
[ANSWER] At the reviewer's suggestion, I have revised the last sentence to make it not so long.
Line 17-20
... At present, there is much research evidence that
blood flow-restricted low-load resistance training (20%‒
30% of one repetition maximum [1RM]) has been reported as a sarcopenia countermeasure in older adults.
Therefore, this training method may be particularly effective in preventing sarcopenia.
< 3 >
Figure 1. Explain the abbreviations.
[ANSWER] According to the suggestions of reviewer1 and reviewer2, Figure 1 (description about 1RM)
was revised as manunscript.
< 4 >
Table 1. Provide references.
[ANSWER] According to the suggestions of reviewer1 and reviewer2, Table 1 was revised as manuscript.
< 5 >
Table 2. Explain the abbreviations.
[ANSWER] According to the suggestions of reviewer, Table 2 was revised as manuscript.
< 6 >
Line 137 and elsewhere: multiple sclerosis is not an epidemiologic problem of older adults
[ANSWER] According to the suggestions of reviewer, the description of “multiple sclerosis” was deleted as
4
follows.
Line 241
...low-load resistance training (e.g., hip/knee arthritis patients, multiple sclerosis patients).
...
<7>
Lines 139/140 and 143/144: redundant repetition
[ANSWER] A reviewer pointed out another revision, and as a result, this section has been removed.
< 8 >
Lines 184/185, 265, 314: Table 1 shows something different
[ANSWER] As pointed out by the reviewer, the description of “Table 1” was deleted. In addition, the
reviewer pointed out another point, which was revised, and as a result, it was deleted in one section.
Line 200
... similar to H-RT, regardless of age [9,73] (Table 1).
Line 237
... muscle hypertrophy in older adults (Table 1).
< 9 >
Line 304: Citations do not correspond to the text
[ANSWER] A reviewer pointed out another revision, and as a result, this section has been deleted.
Round 2
Reviewer 1 Report
The author did an excellent job and considerably improved the manuscript.
Author Response
Dear Reviewer
Thank you for your response to the revised manuscript.
Reviewer 2 Report
Review 1
Various methods of resistance training for prevention and treatment of sarcopenia
Presentation of the manuscript has been considerably improved.
Nevertheless, the description of various methods of resistance training for prevention and treatment of sarcopenia is still not complete. Especially, the effects of high-speed power training should be described.
Or alternatively, the title should be changed to:
“Selected methods of resistance training for prevention and treatment of sarcopenia”
and the fact of selective description should be clearly stated in the limitations.
Likewise, safety and adverse effects (e.g., falls, exacerbation of osteoarthritis) have not been adequately describe.
Specific comments
Line 311-313: redundant repetitions
Author Response
Review 1
Various methods of resistance training for prevention and treatment of sarcopenia
(1) Presentation of the manuscript has been considerably improved.
Nevertheless, the description of various methods of resistance training for prevention and treatment of sarcopenia is still not complete. Especially, the effects of high-speed power training should be described.
Or alternatively, the title should be changed to:
“Selected methods of resistance training for prevention and treatment of sarcopenia”
and the fact of selective description should be clearly stated in the limitations.
[ANSWER] Thank you for the suggestion. The reviewer pointed out that this review is limited to selective training programs. Also, given that this review focuses only on external loads, I decided to include the description of power training in the Limitation section and to "change the title" and "clearly state the fact of the selective description in the Limitation section" as suggested by the reviewer.
Line 2 (Title)
Selected methods of resistance training for prevention and treatment of sarcopenia
Line 363-365 (6. Limitations)
There are several limitations to this study. First, this study focused on resistance training load intensity and discussed selected four training programs. Therefore, although there have been reports of muscle hypertrophy from training using walking, electrical stimulation and high-speed power training, this study did not address them. …
(2) Likewise, safety and adverse effects (e.g., falls, exacerbation of osteoarthritis) have not been adequately describe.
[ANSWER] Thank you for the suggestion. Based on the reviewers' comments, I decided to add the following safety and side effects for the four training methods described in this manuscript.
- H-RT
Few studies have reported the effect of H-RT on the risk of falls in the elderly. However, because H-RT places large compressive loads on joints and pain is a concern, low-to-moderate load resistance training is recommended for patients with knee osteoarthritis and other conditions. Therefore, sentence was added as follows.
Line 144-146
… diabetes [54-56], etc. In addition, H-RT is known to cause joint pain due to high compressive loads [57,58], and low-to-moderate-load resistance training is recommended [59,60]. Thus, although …
References
- Lees, F.D.; Clarkr, P.G.; Nigg, C.R.; Newman, P. Barriers to exercise behavior among older adults: a focus-group study. Aging Phys. Act. 2005, 13, 23-33.
- Leveille, S.G.; Fried, L.P.; McMullen, W.; Guralnik, J.M. Advancing the taxonomy of disability in older adults. Gerontol. A Biol. Sci. Med. Sci.2004, 59, 86-93.
- American Geriatrics Society Panel on Exercise and Osteoarthritis. Exercise prescription for older adults with osteoarthritis pain: consensus practice recommendations. A supplement to the AGS Clinical Practice Guidelines on the management of chronic pain in older adults. Am. Geriatr. Soc. 2001, 49, 808-823.
- Ettinger, W.H. Jr.; Burns, R.; Messier, S.P.; Applegate, W.; Rejeski, W.J.; Morgan, T.; Shumaker, S.; Berry, M.J.; O'Toole, M.; Monu, J.; Craven, T. A randomized trial comparing aerobic exercise and resistance exercise with a health education program in older adults with knee osteoarthritis. The Fitness Arthritis and Seniors Trial (FAST). JAMA. 1997, 277, 25-31.
(ii) L-BFR
Several L-BFR studies have reported on fall risk and patients with knee osteoarthritis. In a systematic review, Gronlund et al. (2020) noted that "L-BFR studies have yet to show a sufficient effect on falls in older adults," but that there is evidence that "L-BFR improves physical performance, muscle strength, and balance capacity, which are associated with fall risk. The report indicates that there are Buford et al. (2015) and Pitsillides et al. (2021), in an RCT study or RCT review, found that "L-BFR is likely to improve muscle strength and reduce knee osteoarthritis pain in older patients with knee osteoarthritis. Therefore, sentences were added as follows.
Line 231-236
… no severe side effects. L-BFR has been reported to improve factors related to falls, such as physical performance and muscle strength [9], although research results on the risk of falls in the elderly are not yet sufficient [83]. In addition, for older adults with knee osteoarthritis, it has been reported to reduce pain to the joints and increase muscle strength, rather than side effects [84,85]. Therefore, L-BFR is expected to have low side effects, such as falls and exacerbation of knee osteoarthritis. Besides, its safety …
References
- Gronlund, C.; Christoffersen, K.S.; Thomsen, K.; Masud, T.; Jepsen, D.B.; Ryg, J. Effect of blood-flow restriction exercise on falls and fall related risk factors in older adults 60 years or above: a systematic review. J. Musculoskelet. Neuronal. Interact. 2020, 20, 513-525.
- Buford, T.W.; Fillingim, R.B.; Manini, T.M.; Sibille, K.T.; Vincent, K.R.; Wu, S.S. Kaatsu training to enhance physical function of older adults with knee osteoarthritis: Design of a randomized controlled trial. Contemp. Clin. Trials. 2015, 43, 217-222.
- Pitsillides, A.; Stasinopoulos, D.; Mamais, I. Blood flow restriction training in patients with knee osteoarthritis: Systematic review of randomized controlled trials. J. Bodyw. Mov. Ther. 2021, 27, 477-486.
(iii) L-ST
In L-ST, there are no reports of studies examining the effects on fall risk or knee osteoarthritis. Therefore, the following descriptions have been added.
Line 293-295
… after 12 weeks of training. However, there are no reports of studies on the effects of L-ST on fall risk or exacerbation of knee osteoarthritis in the older adults, and research on the safety and side effects of this training method is not well evidenced.
(iii) L-ST and L-FAIL
In L-FAIL, there are no reports of studies examining the effects on fall risk or knee osteoarthritis. Therefore, the following descriptions have been added.
Line 349-351
There are no reports of studies on the effects of L-FAIL on fall risk or exacerbation of knee osteoarthritis in the older adults, and research on the safety and side effects of this training method is not well evidenced.
Specific comments
(3) Line 311-313: redundant repetitions
[ANSWER] Thank you for the suggestion. According to the reviewer’s suggestion, the sentences was deleted as below.
Line 319-320
… young adults. Therefore, L-FAIL is likely to be an effective way to promote MPS and muscle enlargement with or without BFR. Thus, there is a high probability that L-FAIL is an effective way to promote MPS and muscle hypertrophy with or without BFR. Most studies using …